# Study protocol for a longitudinal study evaluating the impact of rape on women's health and their use of health services in South Africa

Naeemah Abrahams,[1] Soraya Seedat,[2] Carl Lombard,[3] Andre P Kengne,[4]
Bronwyn Myers,[5] Alesha Sewnath,[1] Shibe Mhlongo,[1] Gita Ramjee,[6]
Nasheeta Peer,[7] Claudia Garcia-Moreno,[8] Rachel Jewkes[1]

For numbered affiliations see end of article.

**Correspondence to**
Dr Naeemah Abrahams;
nabraham@mrc.ac.za

## ABSTRACT

**Introduction** South Africa is a country known for its high levels of HIV infection and sexual violence. Although the interface between gender-based violence, HIV and mental health has been described, there are substantial gaps in knowledge of the medium-term and long-term health impact. The 2010 Global Burden of Disease study excluded many health outcomes associated with rape and other forms of gender-based violence because systematic reviews revealed huge gaps in data and poor evidence of health effects. This study aims to describe the incidence and attributable burden of physical and mental health problems (including HIV acquisition) in adult women over a 2-year postrape period, through comparison with a cohort of women who have not been raped. The study will substantially advance our understanding of the impact of rape and will generate robust data to assist in the development of postrape health services and the delivery of evidence-based care.

**Methods and analysis** This longitudinal study seeks to recruit 1008 rape-exposed and 1008 rape non-exposed women. Women were recruited from health services, and assessments were carried out at baseline, 3, 6, 9, 12, 18 and 24 months. Outcome measures include exposure to risk factors; mental health status; cardio-metabolic risks; and biomarkers for HIV, sexually transmitted infections, pregnancy and stress. The primary analysis will be to compare HIV incidence in the two groups using log-rank tests. Appropriate models to predict health outcomes over time will also be applied.

**Ethics and dissemination** The South African Medical Research Council's Ethics Committee approved the study. As rape is a key element of the study, the safety and protection of participants guides the research process. We will adopt a research uptake strategy to ensure dissemination to policy makers, service providers and advocacy groups. Peer-reviewed journal articles will be published.

## BACKGROUND

Lifetime exposure to rape and other forms of sexual violence is relatively high in most countries, as reported in global prevalence

### Strengths and limitations of the study

► The study will build knowledge in an area with many gaps and will contribute to describing the health burden of gender-based violence.
► The study uses a longitudinal design with a large sample and combines biological and behavioural data.
► The study is conducted in a developing setting.
► Recruiting only rape survivors who report rape means that our study findings will not be generalisable to all rape survivors.
► Sexual violence is often under-reported and may not be revealed by participants in the non-exposed group at screening and enrolment.

studies released in June 2013.[1 2] Although aspects of the health impact of rape have been described, the data available have major limitations. Systematic reviews to establish the magnitude of health effects associated with exposure to physical and sexual intimate partner violence (IPV) and non-partner sexual violence, including rape, indicate that the current evidence is overwhelmingly based on cross-sectional studies and/or clinical populations.[2] Furthermore, the conditions studied are mostly limited to mental ill-health and aspects of reproductive health. It is thus difficult to quantify the health burden in the absence of high-quality outcome data from longitudinal studies.

The prevention of postrape HIV acquisition has been one of the main targets of healthcare following a rape.[3] However, the pathways to postrape HIV acquisition and health outcomes have yet to be fully elucidated. While HIV can be acquired during a rape event, rape may indirectly elevate HIV acquisition risk in the medium and long term through other possible pathways. Evidence

of these pathways stems from research on IPV and HIV.[4] One pathway is via acquiescence, where exposure to IPV reduces women's ability to assert themselves in sexual contexts and determine the timing and circumstances of sex, including condom use and the insistence on fidelity.[5 6] The second pathway is through elevated high-risk sexual behaviour on the women's part, including engagement in sex work, transactional sex, seeking older partners and having multiple partners.[6–8] The third pathway is that while women themselves may not change their behaviour, they may sustain or be drawn to relationships with men who are at higher risk of being HIV positive than other men. There is evidence that men who use violence towards women, including rape, are riskier, have more partners, have more sexually transmitted infections (STIs) and are more likely to be HIV positive. Similarly, men are more likely to partner more vulnerable women.[9–12]

It is likely that these pathways are important postrape and that psychological reaction to rape and rape stigma are powerful mediators of these pathways. These are, in turn, influenced by the circumstances of the rape. There is some evidence that self-blame results in greater and more enduring psychological impact and poorer post-traumatic stress disorder (PTSD) treatment responses.[13 14] Evidence from rape counselling services suggests that court processes after rape may be particularly stressful for women,[15] but the health impact in general has not been empirically described. Research is needed to test these pathways described above in relation to rape exposure to enable a better understanding of their relative importance. This is critical for developing tailored HIV prevention interventions. Furthermore, there is a need for research that combines clinical assessments of postrape mental health (ie, PTSD and depression) with biological markers of stress. The prolonged release of cortisol in the body has been linked to several psychiatric disorders, including PTSD,[16] and it has been suggested that this itself may impact immune system functioning and create enhanced susceptibility to HIV infection. Further research is needed to investigate links between rape, mental ill-health and dysfunctional hypothalamic–pituitary–adrenal axis activity.[17 18]

Evolving research is contributing to the better understanding of the role of genetics in psychiatric disorders, including responses to trauma, such as sexual violence. Psychiatric genetics has the potential to inform our understanding of neuropsychiatric disorders,[19] including the development of PTSD and other postrape mental health outcomes, and to understand why not all survivors develop mental health disorders.[20 21] Factors other than the traumatic event are likely to contribute to their development. Environmental influences, such as trauma (rape), can trigger epigenetic changes resulting in genes being silenced or expressed.[22] Epigenetic mechanisms can therefore be a mediator between genes and the environment. Research is needed to deepen our knowledge of the role of epigenetics in the development of postrape mental health disorders.

The relationship between non-communicable diseases (such as cardio-metabolic diseases) and rape has not been described in longitudinal studies in the literature. Suggestions of a potential relationship can be gained from cross-sectional research on the associations between physical and psychological IPV and self-reported chronic diseases including hypertension, diabetes, asthma, heart disease, cancer and arthritis.[23] It has been suggested that the stress response following rape may lead to the development of chronic diseases and its risk factors via similar causal pathways as in individuals exposed to general psychological stress that are known mediators of poor healthcare.[24] Direct pathways have been hypothesised via physiological mechanisms when traumatic stressors activate a cascade of physiological responses that affect the cardiovascular, metabolic, neural, endocrine and immune systems.[24–26] These include pathways via behavioural coping mechanisms such as detrimental lifestyle choices. Research is needed to understand these associations and the potential pathways using biological markers. Many studies, using different study designs and approaches, have examined the relationship between psychosocial stress and cardiovascular disease (CVD) and its risk factors.[27–29] None of these studies have described the relationship between psychosocial stress triggered by rape exposure and the development of CVD risk factors in longitudinal studies.

In sub-Saharan Africa, an area with a high population prevalence of HIV infection, it is common for women who are HIV positive to be raped. Not much is known about HIV-positive women who are raped and the specific health burden associated with such exposure. In recent years, there has been a substantial increase in HIV testing across sub-Saharan Africa, especially among women attending antenatal services, and a much greater linkage to HIV treatment services.[30 31] High levels of medication adherence and retention during treatment are essential for the effectiveness of antiretroviral therapy (ART) in preventing HIV disease progression and mortality. Consequently, research on barriers to adherence and retention in HIV treatment and care services has been increasing.[32] Psycho-social factors are one group of factors that undermine linkage and retention in care, and among rape survivors, mental illness and stigma are of particular concern because they are recognised barriers to HIV adherence and retention in care.[33 34] Understanding and developing interventions to respond appropriately to the impact of rape on women's ability to adhere to treatment and remain in care is important for optimising women's HIV treatment outcomes.

Given the state of the knowledge on the long-term health effects of gender-based violence, it is not surprising to find few academic papers on recruitment and retention of this vulnerable population. Indeed, longitudinal studies are challenging to conduct among all study populations. The gender-based violence studies that follow participants over a period are mainly from clinical settings, in particular for management of PTSD in populations such as war

veterans, or from evaluations of batterers programmes; none were found among community samples in developing settings.[35–39]

To fill these gaps in the knowledge, we are undertaking research that has the overall goal of advancing the understanding of the health consequences of rape, notably HIV and mental health. This study is particularly well suited to being conducted in South Africa because it is a country with high sexual violence and HIV incidence[9 40] and is the country with the largest population of individuals living with HIV.[41]

## METHODS/DESIGN
### Research aims
The study has one primary aim and eight secondary aims.

### Primary aim
To determine the incidence and attributable burden of HIV acquisition in adult women up to 24 months postrape in comparison to women who have not been raped.

### Secondary aims
1. To determine the incidence, attributable burden and recovery rates of physical and mental health problems that may enhance HIV risk at 3, 6, 9, 12, 18 and 24 months.
2. To determine the individual, relational, social and criminal justice risk factors for health problems at the different time points that may be HIV risk factors and associated with the persistence of symptoms.
3. To estimate the relative importance of the different hypothesised pathways to HIV acquisition.
4. To determine the impact of rape on HIV positive survivors' ability to link to HIV care, retention of treatment and sexual risk-taking behaviour.
5. To evaluate changes in cortisol levels and other steroid hormones (cortisone, testosterone, progesterone, corticosterone and dehydroepiandrosterone) over time, measuring their levels in hair, and to evaluate the stress response of women before and after the traumatic event of rape.
6. To investigate genetic and epigenetic factors as participating biomarkers in the aetiology and trajectory of PTSD among rape-exposed women.
7. To determine the incidence and attributable burden of CVD risk factors, markers of increased CVD risk and hypertension-related and diabetes-related renal dysfunction, comparing rape-exposed and non-exposed women.
8. To explore, qualitatively, the experiences of both rape-exposed and rape non-exposed women and to focus on motivations and experiences of the research process, retention issues and the support provided to them.

The study is called the Rape Impact Cohort Evaluation (RICE) study and is referred as this throughout this paper. This comparative cohort study includes a 24-month follow-up. The sample size calculation addresses the primary aim. This is based on a log-rank test comparing the HIV incidence between rape-exposed and non-exposed groups. To achieve 90% power to detect a significant difference in HIV incidence, assuming an annual incidence of 9% and 6% among the rape-exposed and non-exposed groups, respectively, and given an expected 36-month accrual and a minimum follow-up of 12 months, at a 5% significance level and 10% dropout, 193 HIV positive incident cases will be required.

A minimum HIV negative cohort of 780 participants is required in each of the exposure groups. We will recruit at least 844 HIV-negative participants in each exposure group and the last participant recruited will be followed up for 12 months. Research in KwaZulu-Natal (KZN), where this study is being undertaken, has found an average annual HIV incidence of 10% among young women participating in trials.[42] We anticipate that 20% of participants will be HIV positive at baseline, and as such, a total sample size of 2016 (1008 per arm) will be recruited. We anticipate a 10% dropout based on previous experiences with cohorts in KZN, where 95%–97% retention was achieved in five longitudinal studies.[43] All participants will have data collected at baseline, 3, 6, 9 and 12 months follow-up. Participants recruited earlier in the study will also have data collected post 12 months, at 6-month intervals up to 24 months, which will give greater power to the study.

### Setting
The study setting is in and around the city of Durban in the KZN province. The rape-exposed participants are recruited from four rape services that are spread across the city of Durban. Three of the rape centres are Thuthuzela Care Centres (TCCs). These are one-stop dedicated sexual assault services based at public hospitals providing 24 hours integrated care to rape survivors, including access to police, counselling and medical care, which are also described in the law.[44 45] The health services provided include provision of initial medical and forensic care, and HIV prevention is a strong component of the TCC model of care. The three TTCs from which we recruit our rape-exposed group are based at regional hospitals in the north and south of the city (names omitted to ensure confidentiality). The fourth recruitment site is the Crisis Centre at a regional hospital situated in the city centre. The Crisis Centre is not an exclusive sexual assault service as the TCCs are but provides similar care to rape survivors. Although more TCCs are located at district hospitals in the rural areas surrounding Durban, it would have placed a great financial strain on the study to support rural participants. Also, retention is a known challenge, as reported in HIV research, if study participants are from highly mobile populations, and if the study clinic is situated some distance from where people live.[46] We were lucky to set up our research study clinic (RICE clinic) on a hospital complex in one of the big southern townships, about 20 km south of Durban city centre and within easy

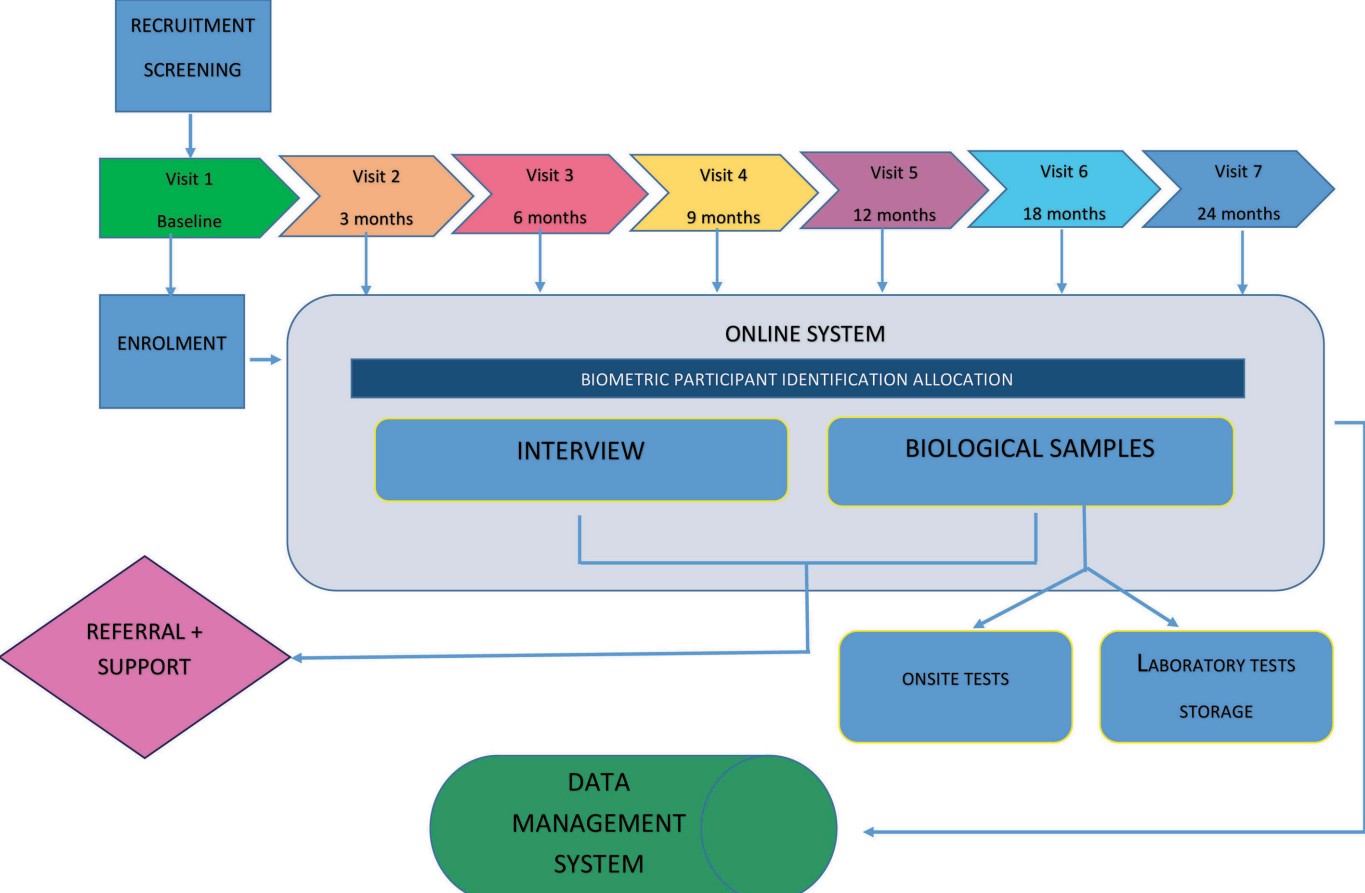

**Figure 1** Rape Impact Cohort Evaluation study flow chart of participants and samples.

access to public transport. We opted to recruit only from services within a 50 km radius of the study clinic. However, survivors from nearby rural areas may seek services at any of the four recruitment sites, and we will recruit these people if they indicate their willingness to travel to the RICE clinic. Non-rape exposed participants are recruited from public health family planning clinics in the same vicinity as the recruitment sites of the rape-exposed women. Permission to recruit at the TCCs was granted by the National Prosecuting Authority (NPA), and the Provincial Department of Health granted permission to recruit at family planning clinics.

At recruitment, we invite women to our RICE clinic where formal informed consent procedures are conducted as well as the baseline interview and assessments (see figure 1). We explain in detail the study procedures and the follow-up clinic visits and offer transport cost or a transport service to and from the clinic.

### Inclusion and exclusion criteria

Rape-exposed participants are women aged 16–40 years who have lodged a complaint of rape within the previous 20 days at a rape centre. Children under 16 years are excluded because they constitute a vulnerable group that requires additional ethical safeguards. The upper age limit of 40 was chosen because the HIV incidence in women is much lower over the age of 40 years. The limit

of 20 days was chosen to ensure that the time points for data collection are reasonably standardised across the study population and to allow for baseline documentation of acute stress reactions (PTSD for the current event can only be assessed at the follow-up visits) and HIV status. Participants are excluded if they are very severely mentally distressed, mentally disabled or more than 14 weeks pregnant.

Participants attending family planning (non-rape exposed) are matched for age (+/−1 year) with recruited raped-exposed women. The choice of family planning clinic clients provides a sexually active control population who, like the women seeking services for rape, are also engaged with health services. Women who report lifetime exposure to sexual violence, as determined by a screening questionnaire, are excluded from the non-exposure cohort. The same exclusion criteria, as for rape-exposed women, such as mentally disabled and more than 14 weeks pregnant, are also applied. We follow the model of recruitment that has been successful in previous studies at sexual assault services in South Africa. These include a series of information giving and consent processes described below and also described in detail elsewhere in other publications.[47 48]

Women recruited in our non-exposed group who report a rape and meet the inclusion criteria, that is, enrolled

within 20 days of the rape, will be switched to the exposed group.

## Assessment of exposure and outcomes measures

Both cohort groups attend the same number of follow-up visits and have the same assessments. The interview content and outcome measures are listed in table 1. Baseline interviews are conducted face to face in the language that participants are comfortable with (Zulu, English). Responses to questions at follow-up interviews are interviewer administered or self-completed, depending on the literacy levels of participants and their familiarity with computers.

Data are being collected in real time using an electronic data management system (see figure 1). Trained staff administer the mental health diagnostic assessment, the MINI DSM-IV and diagnostic interview face to face.[49] The research nurse conducts the clinical assessments, collects the biological specimens (table 2), conducts HIV testing and counselling and administers a pregnancy test and an STI test (self-administered vulvar swab for Trichomonas, which is treated if positive). We follow the WHO recommendations for HIV rapid testing by conducting a second test on all positives as well as an ELISA test (Vironostika HIV1/2 Elisa).[50] Standard pretest and post-test counselling is delivered in a culturally sensitive manner. Specimens for herpes simplex virus type 2 (HSV2) testing are conducted at 6-month intervals from participants who are HSV2 negative at baseline. CD4 count and viral load are measured every 6 months, as is the routine in public sector clinics, in order to not overly influence health-seeking behaviour by observation.

## Follow-up interviews and retention

Participants are given a return date and telephonic contact is maintained between interviews. The study team understands that rape survivors' lives are often in flux in the immediate period after rape, and this can lead to considerable mobility and a change in contact details. The study retention team (retention counsellor, data manager and research assistants responsible for home visits) are primarily responsible for problem-solving retention barriers. However, all study staff are cognisant that their interactions with participants can influence retention. A tracking system to monitor retention has been developed and is updated daily. This allows for immediate intervention (see discussion below on retention challenges).

## Data management and analysis

We use a biometric system to identify, track and link participants and their different data sources (ie, online interviews, clinical data and biological samples) across visits. A unique barcoded participating identification (PID) number is generated at enrolment, and we use barcodes on all samples and documents. Data are collected and entered in real time on a dedicated website and monitored daily by the data manager. Data are downloaded to a secure server and kept in a password-protected system.

Laboratory results are merged into the main dataset using the unique PID.

The primary analysis is to compare HIV incidence in the two groups. The log-rank test will be used to compare the groups, taking into account the interval censoring of event times that will occur due to the visit schedules. The Wilcoxon-type test will be used. To quantify the risk of infection, a Cox proportional hazards model will be fitted and the HR and 95% CI estimated. Interval censoring will also be taken into account in this inference. Mixed-effects regression models will be used to model the mental health status of participants over time and for the comparison of the mean profiles between exposure groups. A random-effects model for participants will be used to account for the clustering effect of the repeated measures. Mixed-effects models, restricted to the rape-exposed group, will be used to evaluate the association of the individual, relational, social and criminal justice factors on mental health, and similar analyses will be conducted for different groups of interest (ie, the HIV-positive group). We will also examine a cascade model depicting the different pathways with the cohort of women who are HIV positive and have been raped and compare it with HIV-positive women who have not been raped. We will handle missing data following the guidelines for clinical studies.[51] Our primary outcome is time to HIV infection. For this analysis, everyone at risk in both cohorts will be part of the analysis, and if someone leaves the study, the time will be censored. Thus, time-to-event analysis accommodates missing data in a natural way. The one assumption is that the dropout and follow-up time will be the same in both groups. For other outcomes, that is, mental health status, the repeated measurements will include the baseline value and the follow-up values. Again, all the information up to the point of leaving the study will be included in the analysis. The estimation will be conducted using maximum likelihood as described above. This is a different approach to accommodating missing data than multiple imputation.

It is possible that the two groups are different for important variables, which may also impact on risk for sexual violence and HIV acquisition. The control of confounding variables is therefore critical in our study. We collect a wide range of socio-economic variables and known confounding variables (see table 1). Furthermore, we minimise confounding by choosing a control group that also uses health services, and we recruit from health services in the same area.

## STUDY PROGRESS AND CHALLENGES

A pilot study to assess the feasibility of recruitment and administering the assessment schedule was conducted between June and October 2014. We have made a few changes to the initial protocol, mainly the change of the recruitment window period and the addition of recruitment sites, children and substudies (addition of genetics, epigenetics, cardiovascular samples and the qualitative component). Recruitment and enrolment commenced

**Table 1** Exposure and outcome measures at each data collection time point

| | Variable group | Measures | Source of measures |
|---|---|---|---|
| A | IPV exposure | Ever physical, sexual, emotional<br>Past year, postrape | WHO multi-country study instrument[61] (tested in South Africa by Jewkes et al)[62] |
| B | Rape circumstances and physical trauma/ injuries | Rape circumstances: number of perpetrators, relationship to perpetrator, weapon use, degree of violence, type of rape<br>Intimidation by the perpetrator or friends | Jewkes et al[63]<br>*(Questions to be asked after criminal case have been closed)* |
| C | Rape stigma and social support | Rape stigma and self-blame<br>Coping mechanisms<br>Rape disclosure<br>Social support<br>Social disruption: leaving home, school, relationship | Social Support Appraisal Scale[64]; Brief COPE[65]; South African Stigma Scale[66] |
| D | Demographic factors | Demographic factors: age, income, education, employment, relationship status, sexual orientation | Jewkes et al[62] |
| E | Mental health outcomes | Depression<br>Anxiety disorders<br>PTSD<br>Suicidality<br>Other anxiety disorders<br>Alcohol and drug abuse<br>Cortisol level in hair<br>Epigenetic changes<br>Telomere length | MINI for psychosis and anxiety disorders[49]; CES-D (depression)[67]; AUDIT (alcohol)[68]; DUDIT (drugs)[68–70]; Life Events Checklist[71]; Davidson Trauma Scale (PTSD)[72 73]<br>Cortisol in hair and epigenetics analysis from blood samples |
| F | Other trauma | Previous trauma: child abuse, other interpersonal violence or traumatic life events<br>*Follow-up interview: rape, interpersonal violence or traumatic life events since the last interview* | Childhood trauma scale Bernstein et al[74] modified for South Africa Jewkes et al[75]<br>Life Events Checklist; rape and forced sex measures from Jewkes et al[62] |
| G | Criminal justice processes and circumstances of the rape | Case progression: case dropped, arrest, bail, guilty plea, case withdrawn, trial commenced, testimony in court given, trial outcome<br>Perceived satisfaction with the system including information given about whereabouts of accused | Jewkes et al[63]; satisfaction questions to be developed<br>*(These questions will be asked once criminal case have been closed.)* |
| H | Acquiescence to male domination of sex | Attitudes to gender relations and perceived social norms<br>Consistent condom use<br>Relationship control scale<br>Sexual frequency | Jewkes et al[62]; attitudes developed in South Africa and validated in Jewkes et al[76] |
| I | Own sexual risk behaviour | Transactional sex<br>Partner numbers ever and past year<br>Concurrency | Jewkes et al[62 75] |
| J | Partner riskiness | STIs of current partner<br>Use of IPV by current partner<br>Partners' concurrency/polygamy | Jewkes et al[62 4] |
| K | Biological risk enhancers or co-factors | Hormonal contraception use at baseline and ongoing<br>Pregnancy loss (since last interview): termination, miscarriage | Questions from the Demographic and Health Survey[77] |
| L | Cardio-metabolic risk assessment | Screening questions for risk assessment<br>Measurements: hip and waist, BP, BMI, glucose status (glycaemia/Hba1c) and lipid panels<br>CRP ultrasensitive (cardiac), Gamma-GT, ALT (SGPT), AST (SGOT), creatinine (serum), | Questions from the WHO STEPs questionnaire[78] |

Continued

**Table 1** Continued

| | Variable group | Measures | Source of measures |
|---|---|---|---|
| M | Healthcare seeking practices postrape health, HIV testing and adherence | Completion of PEP and STI treatment Emergency contraception HIV testing postrape and previously HIV disclosure Attendance for health assessment Commencement of ART and adherence Experience of HIV-related ill-health Satisfaction with healthcare provision | Items developed and tested in pilot |
| O | Reproductive and child health | Contraception, fertility, pregnancies, reproductive health Child health including babies born during RICE study participation | Questions developed and adapted from Demographic and Health Survey |

ALT, alanine aminotransferase; ART, antiretroviral therapy; AST, aspartate aminotransferase; AUDIT, Alcohol Use Disorders Identification Test; BMI, body mass index; BP, blood pressure; CES-D, Center for Epidemiologic Studies Depression Scale; CRP, C-reactive protein; DUDIT, Drug Use Disorders Identification Test; IPV, intimate partner violence; MINI, Mini International Neuropsychiatric Interview; PEP, postexposure prophylaxis; PTSD, post-traumatic stress disorder; RICE, Rape Impact Cohort Evaluation; SGOT, serum glutamic oxaloacetic transaminase; SGPT, serum glutamic pyruvic transaminase; STI, sexually transmitted infection.

in November 2014. By April 2017, we had achieved 56.9% of the recruitment target for the rape cohort (n=574) and a slighter better rate among the non-exposed cohort (64.2%; n=647). Our recruitment rate per month among the rape-exposed cohort improved from 13 per month in 2015 to 23 per month in 2016, and this was mainly due to interventions discussed below. We expect this to improve marginally and hope to have our full rape cohort enrolled by August 2018. The expected end date of the study is August 2019 when the 12-month interviews of the last participants will be conducted.

### Recruitment and enrolment challenges

For various reasons, our recruitment was much slower than we anticipated in the planning phase. Our initial plan was to enrol participants within 5 days of the

**Table 2** Clinical assessments and biological samples at each visit

| | Baseline | 3 months | 6 months | 9 months | 12 months | 18 months | 24 months |
|---|---|---|---|---|---|---|---|
| Biometrics | X | X | X | X | X | X | X |
| Clinical assessment | X | X | X | X | X | X | X |
| Mental health assessment | X | X | X | X | X | X | X |
| HIV and trauma counselling | X | X | X | X | X | X | X |
| Laboratory evaluations | | | | | | | |
| HIV rapid test and ELISA | X | X | X | X | X | X | X |
| Pregnancy test | X | X | X | X | X | X | X |
| Trichomonas swab | X | X | X | X | X | X | X |
| HSV2 | X | X | X | X | X | X | X |
| Glucose/lipid panels CRP ultrasensitive (cardiac), Gamma-GT, ALT (SGPT), AST (SGOT), creatinine | X | | | X | | | X |
| Cortisol (hair sample) | X | X | X | X | X | | |
| Genetic and epigenetic tests | X | X | X | X | X | X | X |
| Virology (HIV positive) | | | | | | | |
| Viral load and CD4 | X | | X | | X | | X |
| p-Antigen 24 (sero-converted) | X | X | X | X | X | X | X |
| Storage | | | | | | | |
| Plasma | X | X | X | X | X | X | X |
| Serum | X | X | X | X | X | X | X |

ALT, alanine aminotransferase; AST, aspartate aminotransferase; HSV2, herpes simplex virus type 2; SGOT, serum glutamic oxaloacetic transaminase; SGPT, serum glutamic pyruvic transaminase.

rape event, but we found this to be a huge challenge and enrolment was very slow. The window period for enrolment was extended to 10 days and later to 20 days postrape. We cannot enrol beyond 20 days postrape because we assess mental health outcomes, such as PTSD, that are time sensitive to the traumatic event, as well as the assessment of HIV status at baseline, which is a key measurement.

We found that a larger proportion of patients reporting rape to the TCCs are children, which is different to the age profile statistics provided by NPA at the planning phase of the study. We used 2011–2013 NPA data to plan enrolment strategy. We also do not recruit mentally challenged rape survivors or those who are too emotional to discuss research participation.

Our initial recruitment process was dependent on the trained TCC staff's ability to identify eligible participants, introduce the study, gain consent to be contacted by us (the researchers) and finally collect accurate contact details. The project staff then contacted the potential participant and invited them to our study clinic for formal consent procedures and the baseline interview (see figure 1). Although TCC staff were keen to assist us, our study was not their priority. We therefore enlisted the Life Line counselling staff, who provide the initial counselling service at the rape centres and we contribute towards their salary. However, when funding (United States Agency for International Development funding) for counselling services at the rape centres decreased at the end of 2015, the RICE investigators decided to employ dedicated RICE recruiters because recruitment remained low and it made financial sense to employ staff to increase the enrolment rather than to enrol over a longer period. We employed two of the Life Line counsellors to cover four recruitment sites. They set up the initial appointments and arrange the collection and transportation of the participants to our research clinic. Frequently, participants become non-contactable after agreeing to attend our research clinic for formal consent procedures, and this is a challenge. Participants provide incorrect contact numbers or our calls and voicemails remain unanswered. The latter often happens when final arrangements are being made or at time of transport pick-up. Such behaviour is also noted among the non-exposed group recruited at family planning clinics, and we suspect potential participants are reluctant to inform us directly that they do not want to participate in the research. They may feel socially coerced to agree but find a way to discontinue contact with us. We know that rape survivors' lives are difficult postrape and they change their phone numbers (phones are sometimes lost/stolen at the time of the rape), or they relocate. We also know that their mental health status interferes in their ability to regain normality.[13][52] This is an inherent challenge in our study, and staff are sensitive to rape survivors and accept that this group of women may not want to participant in our study.

## Retention challenges and responses

Retaining participants in longitudinal studies is challenging, and rape survivors bring additional concerns. Indeed, abused women have been identified as among the 'hard to reach' populations in research, similar to drug users.[38] Retention is also not often discussed in peer-reviewed journal articles, where study findings are the focus. We searched for lessons learnt from other studies but could not find longitudinal studies with large rape cohorts. Similarly, the HIV/AIDS sector has always grappled with retention in cohort studies.[46][53] As indicated above, we have participants who relocate to other provinces for safety reasons or as a coping strategy, or they find jobs in other towns. We are aware survivors experience a range of emotional difficulties with varying levels of distress and coping skills, and we know receiving support and care from others assist them on their road to recovery. We also know that there is poor integration of mental health within rape services, and although women can return to the rape centre, they often choose not to. Our study has a full-time qualified trauma counsellor who provides counselling support at the study clinic. Many, however, do not use this facility, and we may need to understand this better in follow-up studies. We find that women from both cohorts, who are counselled at our clinic, often seek assistance for their daily struggles. Women will report that their stresses are related to their difficult lives such as trying to find a job and securing income. Counselling sessions are often about assisting them to cope with these hardships. We expected more women to withdraw from the study because the research reminded them of the incident, but to date, among the 26 women who withdrew at the time of writing this paper, only 2 mentioned this as the reason. There may be others who have withdrawn, but because this has not yet been communicated to us, we consider them loss to follow-up (LTFU).

We have activities beyond reimbursement to support retention. These include updating locator information at each visit, consent to use social media, maintaining telephonic contacts between clinic visits, individual birthday wishes via WhatsApp, offering transport from the rape centre and a Saturday morning clinic to accommodate students and working women. We try to provide women with a pleasant experience at the clinic and offer manicures and nail painting. We constantly explore other ways to improve their experience and are currently exploring offering 10–15 min head and neck massages. At the end of 2015, our retention rate was between 50% and 75% (depending on the visits), and we intensified our retention activities through an incremental increase in reimbursement monies for each additional visit, a present (necklace/bag/earrings) for those who attend their 12-month and 24-month visit and an additional snack of a small chocolate or a packet of peanuts. We employed a retention counsellor whose primary task is to track and monitor participant attendance and to conduct problem-solving counselling of the retention barriers that

are identified. We also employed a driver for home visits, initially for once a week, which then increased this to twice a week. In January 2017, we employed two drivers. One was dedicated to conduct daily home visits to those who missed a visit, and the other was employed to transport participants. However, home visits are not always successful. Sometimes, a home address is not found or the participant is not known by the household. This may be deliberate misinformation given to us as a form of avoidance. We also find cases where participants have left their rented homes and contact is completely lost. Nonetheless, many home visits result in finding participants or getting updated telephone numbers.

We developed an early warning system of missed attendance so that home visits are carried out soon after a missed appointment. These retention and tracking activities have become more complex because the number of baseline participants and follow-up visits increase. Each of our 2016 participants are expected to attend seven times (a total of 14 056 clinic visits). We started a monthly raffle where a grocery or pamper parcel worth R150.00 (US$10) can be won. Participants enter their study number into the visit box at each visit, and there is a draw at the end of each month.

Transport to the clinic remains one of the key challenges. We recruit women from a wide geographic area, and our study clinic within a regional hospital complex is only accessible by very unpredictable local taxi services. Participants report spending a long time travelling. Since the start of 2017, we have been offering all women transport to the clinic, and participants are collected in the morning at their homes, the rape centre or a known agreed landmark (police station/taxi rank etc). Some women do not use the collection service because they profit slightly from the transport reimbursement and therefore prefer to use public transport. Transport coordination requires very good communication among staff members and between the staff and participants. This is possible because almost all participants agree to use their personal phones to maintain contact with us and we commonly use Whatsapp—especially when arrangements are finalised after office hours.

### Lessons learnt and limitations

We have learnt many lessons on how to set up and manage a longitudinal study with rape victims, and we continue to learn as our study progresses. We do not expect enrolment of rape-exposed participants to increase much beyond 26 per month because there are few other options left to increase recruitment from these four recruitment sites. Broadening the recruitment to other TCC sites is not financially feasible. We must accept that some women are not assertive enough to tell us that they do not want to participate in our study, and to avoid disapproval, they agree initially and thereafter evade contact. We must also be cognisant that power imbalances between researcher and participant may also be at play. More importantly, the postrape period is very difficult for survivors and they

have to negotiate many unsympathetic environments, which may include the rape services (health, police, justice and social), unsupportive family and friends, as well as a broader social environment wherein rape stigma remains a reality.[13 47]

LTFU is the most common cause of attrition in cohort studies,[46] and the loss of contact with the participants has caused frustration among staff. Continuous support and allowing staff to discuss their frustrations have enabled them to continue interacting sympathetically with trauma victims and to prevent vicarious trauma. The RICE study counsellor supports staff in addition to participants. We also ensure staff have access to an expert psychologist who spends a day with them every quarter to assist them in working with traumatised rape survivors. Staff have said they are encouraged when returning participants are clearly on a healing path (ie, they are less depressed or accept an HIV status and are starting ART), which makes the work a rewarding experience.

Difficulties in retention are well known in cohort studies, and we have learnt much in this regard. In hindsight, we should have employed the retention counsellor earlier in the study and should have placed greater effort on the home visit programme. We are continuously assessing and discussing how to improve retention, and we hope the two recent initiatives (transport service and the raffle) will attract people to return. We may not be aware of all the factors influencing retention, and we hope the data from home visits and from our qualitative study will provide some insight to assist us in future longitudinal studies with sensitive participants. This study will also allow us to explore differential retention and LTFU between the two groups. Mental health may also influence retention of women who are depressed and/or anxious and/or using substances and are reluctant to attend the study clinic. The problem that women may struggle to attend care of any form after rape is well described in the literature on rape research because the attendance reminds them of the trauma.[54 55] This will affect overall retention.

We are aware of the limitation of generalising the findings to all rape survivors because low levels of reporting rape to the police are a well-known phenomenon locally and globally.[15 56] Women who report a rape are likely to be different in several aspects from the wider group of women who experience rape and who do report it. Those who do report are more likely to have experienced injuries, to have had witnesses or to have been raped by non-partners (strangers). However, these are also women who are accessible from the wider population of women who are raped for longitudinal research on health outcomes. Women who come forward to the police or to health services are also the group who requires services and so is an important group to study. From a pragmatic point of view, the methodology employed here is justifiable, especially considering logistical difficulty and the high cost of recruiting a population-based cohort. Generalisability may also be

impacted on by the non-random, non-census nature of recruitment. Women may decline to participate in the study, and this could introduce bias, but we expect this to happen in both groups. We will aim to reduce this bias but will have to be cognisant that participation is by voluntary consent. It is also possible that some women who decline participation may do so because of pre-existing mental health problems. However, our control group is also women who attend health services and, in this way, the two groups are similar.

## ETHICS AND DISSEMINATION

Ethical approval for the study was granted by the South African Medical Research Council's (SAMRC) Ethics Committee. Rape is a key element of our study, and the safety of participants is paramount and guides the research process. Project staff receive extensive training on the ethics, safety and protection of participants. We use lessons learnt from previous studies of participants recruited at rape centres.[47 48] We introduce the study to participants as a *Women's Health and Well-being Study* to protect women from being identified as rape survivors. The study follows the WHO ethical and safety guidelines on conducting research on gender-based violence and applies good clinical practice.[57] We also draw on guidelines developed for conducting intervention research on violence against women.[58] We provide women with sufficient information as required by the SAMRC's Ethics Committee on specific areas such as biometrics, children involvement (see below), genetic samples, HIV testing, hair samples and storage of blood. There is a separate consenting process for each of the above. We developed a brochure and information sheets for participants. Our previous experience highlighted the importance of providing full information, time to reflect and testing the understanding of consent before consent forms are signed. Participants are given a clear explanation of voluntariness of participation, and the option of withdrawal is given at each visit.

Our study includes 16-year-old and 17-year-old girls who are recruited at one recruitment site (Crisis Centre). We did not receive approval from NPA to recruit those under 18 years at the TCCs. For the children, we seek consent from both the parent/legal guardian and the child. It is important that the children we approach understand that their attendance at the rape centre to seek assistance will be revealed to the parent/legal guardian if disclosure has not yet occurred. We are cognisant that some girls may not want their parent/guardians to know about the rape event, and we will not recruit girls if they are opposed to disclosure. The same applies to 16-year-old and 17-year-old children recruited at the family planning clinics. We consider 16 year olds to have sufficient maturity and mental capacity to understand the study (this is acknowledged in the Children's Act No. 38 of 2005 as amended by Act 41 of 2007), and the information sheet for them is the same as those for adult women. We only

enrol a girl if both the parent/legal guardian and the child give consent to participate. We do not seek consent for HIV testing from the parent/legal guardian because according to South African law, parental consent is not required.

The research involves minimal risk to participants. However, questions may be intrusive and trigger emotions that can cause short-term psychological distress. All staff are trained to respond to any distress during the interviews and to provide containment. A trauma counsellor, employed at the start of the study, is available to provide support to participants and staff. Women who require further assistance are referred to appropriate services. NPA trained the staff on what may/may not be discussed with rape survivors to ensure the research does not compromise the legal process. To protect the project staff from having to testify in a trial and the integrity of the legal process, the questionnaire does not ask about circumstances of the rape until the rape case has been closed or the trial has commenced. We expect to have many cases closed either by the survivors or by the investigating team by the end of the follow-up period as shown in research on rape attrition in South Africa.[59] We also do not have a concern for recall bias as such information on the outcome of the legal process is of great importance to most people.

We do not expect any major medical risks as part of the study. Participants may find the prospect of an HIV test distressing, but we anticipate that most participants would have been tested for HIV and would know their HIV status from the rape care received, from antenatal care or from family planning services. In addition, there have been major national HIV testing campaigns across the country.[60] Our research nurses are trained to carry out HIV counselling and, together with the study trauma counsellor, can provide appropriate support and referral.

Project staff monitor adverse events at each visit. A serious adverse event is defined as experiencing violence by a partner due to participation in the study, or injury or hospitalisation due to the violence or death of a study participant for any reason. The research nurse monitors and documents the occurrence of adverse events since the last study visit, at each visit. The nurse follows appropriate procedures for notification and responds to adverse events that arise. The number of events occurring in each cohort will be enumerated to determine whether there is a differential risk between study arms. The data manager will also monitor the adverse event data. Project staff are responsible for determining whether participation in the study is related to the adverse event or not, by exploring the circumstances surrounding the event. All events are reported to the SAMRC's Ethics Committee.

All staff are trained on matters of confidentiality and this is maintained in the retention activities. Contacting participants in the follow-up period poses a particular risk as identified in the guidelines for conducting gender-based

violence longitudinal studies,[58] and we only recruit participants who understand and agree on the follow-up nature of the study. At recruitment, we identify the different ways in which we can contact the participants (short message service, WhatsApp and home visits) and this is reviewed at each visit. All information is stored on secure computers or servers and protected by a firewall and passwords. Study data are kept separate from the forms that can link data to the participants (consent forms and retention forms). Participants are compensated with R80.00 (US$5.50) at the baseline interview with an incremental addition of R20 (US$1.30) at each follow-up visit. Participants are compensated for travel costs or transport is arranged if required. Participants are offered snacks, such as a sandwich and a cool drink, as well a choice of a small chocolate or a packet of nuts, during the visit to the research clinic.

Our study will generate an extensive set of results of interest to a wide audience. We will adopt a research uptake strategy that will involve policy makers, service providers, advocacy groups and academia, that is, peer-reviewed journal articles.

**Author affiliations**
[1]Gender and Health Research Unit, South African Medical Research Council, Tygerberg, South Africa
[2]Anxiety and Stress Disorder Unit, University of Stellenbosch, Cape Town, South Africa
[3]Biostatistics Unit, South Africa Medical Research Council, Cape Town, South Africa
[4]Non-Communicable Disease Research Unit, South African Medical Research Council, Cape Town, South Africa
[5]Alcohol, Tobacco and Other Drug Research Unit, South African Medical Research Council, Cape Town, South Africa
[6]HIV Prevention Research Unit, South African Medical Research Council, Durban, South Africa
[7]Non-Communicable Disease Research Unit, South African Medical Research Council, Durban, South Africa
[8]Department of Reproductive Health and Research, World Health Organisation, Geneva, Switzerland

**Acknowledgements** We thank the South African National Department of Health, the KwaZulu-Natal Department of Health and the National Prosecuting Authority for assisting in accessing the participants. We also thank Life Line and Child Line for their assistance. We also thank the RICE research team for their commitment to the study, and lastly, we thank all the participants for their time and sharing their lives with us.

**Contributors** NA and RJ conceptualised the study and wrote the first draft of the proposal. CL, SS, APK, GR, BM, AS, SM, NP and CG-M contributed to the completion of the study protocol. NA wrote the first draft of this manuscript, and all authors contributed to and approved the final version to be published.

**Funding** This research and the publication thereof is the result of funding provided by the South African Medical Research Council (SAMRC) in terms of the SAMRC's Flagships Awards Project SAMRC-RFA-IFSP-01-2013/ RAPE COHORT.

**Competing interests** None declared.

**Ethics approval** South African Medical Research Council Ethics Committee.

**Provenance and peer review** Not commissioned; externally peer reviewed.

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
