## [Reviewer comments · BMJ Open]

ARTICLE DETAILS

TITLE (PROVISIONAL)	Study protocol for a longitudinal study evaluating the impact of rape on women's health and their use of health services in South Africa
AUTHORS	Abrahams, Naeemah (proxy) (contact); Seedat, Soraya; Lombard, Carl; Kengne, AP; Myers, Bronwyn; Sewnath, Alesha; Mhlongo, Shibe; Ramjee, Gita; Peer, Nasheetta; Garcia-Moreno, Claudia; Jewkes, Rachel

VERSION 1 - REVIEW

REVIEWER	Miriam Hartmann RTI International, South Africa
REVIEW RETURNED	24-Apr-2017

GENERAL COMMENTS	This protocol of a longitudinal study of the impact of rape on women's health provides critical information for the field regarding a novel study approach to explore the topic. The inclusion of challenges and specific solutions utilized to date offer practical knowledge to other researchers interested in implementing similar studies. I have only a few suggestions. One is to include questions developed and tested in an Appendix that are not existing in the literature (e.g. questions in row G and M of Table 1). This will aid in the reproducibility of the study. The second is comment on confidentiality in the context of a longitudinal study, including among contact points via whatsapp. It may be relevant to review the updated WHO guidelines on ethics and safety in intervention research on violence, which comment on follow-up visits. For whatsapp in particular, guidance on whether the phones should be personal or professional and information on norms around deleting messages should be mentioned. Finally, the authors should do a quick review of the spelling and grammar as I noticed a few places where incorrect words were used (e.g. p. 12, line 6 initially should be initial).
---

REVIEWER	Mary Ellsberg Director, Global Women's Institute, George Washington University
REVIEW RETURNED	26-Apr-2017

GENERAL COMMENTS	This is an excellent proposal for a study that is much needed. The longitudinal design will enable the researchers to address issues around the links between rape and HIV infection in a way that has not been possible to date with cross sectional study designs. The study faces considerable challenges both in terms of recruitment, retention but the pilot experience has allowed the researchers to find creative ways to address these concerns. Ethical concerns are also foremost in a study of this type, but the researchers have presented
--

	a very complete plan to prevent and/or mitigate any threats to the physical or emotional safety of participants. This study will make a huge contribution to knowledge about how to best respond to survivors of rape, and to reduce HIV incidence among rape survivors.
--	--

REVIEWER	Hyunkag Cho Michigan State University, USA
REVIEW RETURNED	09-May-2017

GENERAL COMMENTS	This is a wonderful study protocol, well defined and thought out.
---

REVIEWER	Giulia Ferrari London School of Hygiene and Tropical Medicine AND School of Social and Community Medicine (soon to be Bristol Medical School), University of Bristol United Kingdom
REVIEW RETURNED	10-May-2017

GENERAL COMMENTS	This paper presents the protocol for a longitudinal study of the impact of rape on women's health, their use of health services and experiences of violence. It is a very interesting, valuable and very well designed study. It will fill important gaps in our knowledge in this area, and will provide much needed evidence for related research and for public health policy and interventions I offer three types of comments and suggestions. Mostly related to the structure of the paper and study methodology. Structure of the paper It would be useful if the authors strengthened consistency between the research gaps they highlight in the background section, the listed outcomes, the description of the analysis they intend to carry out, and the limitations. Below, I report a numbered list of study aims (for ease of reference) and map a possible correspondence between study aims and the gaps the authors identify in the background section. I would encourage the authors to make these parallels more explicit in the protocol, throughout the sections, and ensure they discuss aims and methods relevant to each gap, and that all gaps and methods are included for each of the listed aims. Primary aim: HIV acquisition + attributable burden, 24 months post rape, compared to a cohort of women with no rape. Secondary:  i) Incidence, attributable burden and recovery rates of phys and mental health problems that may enhance HIV risk between baseline and 24 months. ii) Individual, relational social and criminal justice risk factors for health problems that may be HIV risk factors and associated with symptoms persistence. iii) Relative importance of pathways toward HIV acquisition iv) Impact of rape on HIV+ survivors' ability to link to and stay in care, and sexual risk-taking behaviour;
--

- v) Changes in cortisol and other steroid hormone over time using hair samples, and stress response before and after trauma
- vi) Genetic and epigenetic factors in etiology of PTSD in rape exposed women
- vii) Incidence and attributable burden of CVD factors, markers of increased CVD risk, hypertension and diabetes-related renal dysfunction comparing rape-exposed and non-exposed women
- viii) Qualitative experiences of rape-exposed and non-exposed women,
 - a. & focus on motivations and experiences of the research process, retention issues and the support provided to women

Mapping between gaps/hypotheses and aims

Hypotheses and gaps in the literature	Study aims
HIV may derive from rape directly	primary
OR, it may become more likely via	
i) "Exposure to IPV reduces women's ability to assert themselves sexually, and determine timing and circumstances of sex, including condom use and fidelity;"	Possibly refers to aim number iii, suggesting a structural equation model? Model is not explicitly mentioned in analysis
ii) "Elevated high risk sex behave on the part of women; including engagement in sex work or similar and having multiple partners"	secondary aim number iv
iii) "No change in women's behaviour, but drawn to men with high HIV risk."	secondary aim number iv
"Further research is needed to investigate links	
a. between rape, mental ill-health and dysfunctional hypothalamic-pituitary-adrenal (HPA) axis activity."	secondary aim number i
b. "Research is needed to deepen our knowledge on the role of epigenetics in the development of mental health disorders post-rape."	secondary aim numbers vi and v
c. "No study has described relationship between psychosocial stress triggered by rape exposure and development of cardiovascular risk factors in longitudinal settings."	secondary aim number vii
d. No gap, but mention in study challenges paragraph and elsewhere*	secondary aim number viii
e. No gap, but mention in analysis paragraph	secondary aim number ii
f. "Effect of rape on adherence	secondary aim

	for HIV+ women. Psychosocial factors that hinder women’s adherence to care are understudied. Developing interventions that respond appropriately to the impact of rape to ensure women remain in care is important.”	number iv
*Explaining retention challenges in this (and similar, e.g., IPV survivors?) populations is missing from the background section, but qualitative research to explore these is introduced in the aims (secondary aim number viii) and explained as a later addition to understand challenges (p12, line 28). I would suggest the authors include considerations on these challenges and on the paucity of specific research. Incidentally, there is some on IPV by Gondolf, as the authors are almost certainly aware. As an aide memoir, a lot of it does focus on perpetrators, but see Gondolf and Deemer 2004: though US-focused, this paper may offer some interesting insights. It would be useful if in the analysis section the authors also explained that the study of retention challenges is to be conducted using qualitative methods (only?), and gave some information on which methods they intend to use (data collection methods, sample size and timing of interviews, type of analysis, limitations etc.) throughout the paper. Relatedly, discussion of evidence gaps in women’s experiences (to be investigated with qualitative data – see secondary aim viii) is also not included. Would the authors consider inserting it, similarly to retention issues, to contextualise the qualitative component of the study? Methodological comments The following paragraphs report comments on the methodological issues that arise reading the paper, so they mostly follow the order in which information is presented in the paper. Page 7 Study site/sites: the paper would benefit from a clearer description of the sites. The “Setting” section (page 7, lines 37 to 45) describes the general settings where participants are recruited. However, it does not provide information on the total number of centres recruitment happens at, nor of the urbanicity of the sites. Further, the role of and rationale for the research site is not clearly introduced in this paragraph. Later in the paper, there are mentions of participants being transported to the site for tests. It would aid with clarity if the role of the research site and women’s visits to the site were introduced in the “setting” section More generally, I wonder if the authors would consider providing a bit more detail on the study processes to help the reader understand the study a bit better. The paper contains detailed information on which tests are done at which points in time, and which questions are asked at each point in time, and measurement is clearly discussed. However, some more explanation of the context would help the reader gain a better understanding of the study process. For example, the authors mention “these four sites” on page 14, lines 29-30. However, these have not been introduced beforehand in the text and it is unclear what the authors are referring to. They do		

mention having had to increase the number of “study sites” (page 12, line 27), but this is not necessarily related to the following remark on page 14. Also, there is an explicit mention of (one of) the site(s) (page 14, line 16 – is this the research site? Is this where the research clinic is? Unclear), which the authors might consider turning into a more generic indication of locality to further protect participants’ privacy. They may wish to consider further specifying the function of this site in relation to the others to allow readers to gain a better understanding of the study process, in place of providing its name. Similarly, see the mention of “RICE” on page 10: as noted below, this is not introduced anywhere in the paper. A short explanation of the flow of participants through the study and relevant research actors could prove useful and could be included in this section. For example, in addition to the recruitment and retention strategies described, and of the number and geographical location of the recruitment sites, the authors could also include some information on where and how the various follow up visits are done, and whether this changes between groups; the role of the research site (same as study site?), and who is involved in taking measurements. Perhaps a figure/diagram would be sufficient?

See also:

Page 10

Lines 33-34: RICE staff mentioned for first time. RICE not been previously introduced in paper. RICE study is then mentioned several times, but it is not made clear what/who RICE is. Would the authors consider spending a few words clarifying who RICE are, and what their role in the project is? This becomes indirectly apparent later, at least in part. It would be good to have an introductory succinct description of their role.

And

Page 14

Line 30 “these four sites” unclear. These sites have not been introduced before.

Finally, the authors may want to explain the reason for their choice of recruitment centres for exposed women. This may not be clear to readers not familiar with the South African context, because they may not be aware of the role of the TCCs and the challenges rape survivors face when reporting at police stations. If similar data exist, it would also be useful to provide an indication of the percentage of rapes reported to TCCs versus other services, to get a sense of how representative of rape survivors who report to services the sample may be.

Line 16: tablets indirectly introduced?

Line 55: unclear – is PTSD not collected at baseline? Why not? How are acute stress reactions measured?

Page 8

Inclusion & exclusion criteria, and confounding

Line 3:

I like the strategy of only including women engaged in services this will reduce the impact of behavioural and socio-economic confounders. Matching on age also very good.

However, control women may be different from exposed women on other accounts such as, e.g., socio-economic factors like educational attainment, labour force participation, etc. I would

encourage the authors to discuss these and/or similar confounding variables, and discuss how they are tackling this challenge. Are they collecting data on relevant confounders, if they are not stratifying by these at the data collection stage? How do they think confounders may bias study results (in which direction), and what are they doing to account for them? In the limitations section, the authors are also encouraged to discuss how bias from confounders may affect study results and what analytical strategies they are putting in place to mitigate this.

Line 5: Related to the above point, excluding women with lifetime exposure to sexual violence is justifiable if the authors want to see the effect of rape on HIV incidence on women who are exposed to (recent) violence. However, this choice is likely to reduce the comparability between the two groups, as the women in the exposure group are likely to have been exposed to lifetime violence prior to the incident that triggered their contact with the services. Could the authors elaborate further on this?

In addition, could the authors mention how exposure to sexual violence after recruitment may affect participation in the study for women in the control group?

Line 7: "Other exclusion criteria, as for rape exposed women, are applied." Unclear. Could the authors clarify if all other criteria or a subset are followed?

Measurement

Line 40: AUDIT-C would be sufficient to determine alcohol misuse. Authors may wish to consider using this three-item version of the measure instead of the 10-item AUDIT measure to reduce interviewee burden. The AUDIT-C has been extensively used in South Africa, and has generally shown good psychometric properties and high correlation with the AUDIT measure (see, e.g., Hartley et al. 2011 in *Reproductive Health*)

Page 10

Analysis

"*Data management and analysis*" section: I would encourage the authors to review this section. Regarding the analysis, the authors say they will estimate Cox proportional hazards models, and mixed effects and random effects models for different types of analyses. It would improve the paper's transparency if they mentioned which covariates they are considering including in these models. For a cohort study, in all of these models, confounders due to imbalance and variation in covariates will have an effect on the estimates. It is therefore very important that the authors show they are aware of this, and have been thinking about which covariates should be controlled for in the model, given that only age was included in the matching/stratifying strategies. I would encourage the authors to discuss which confounders they are measuring, and provide indication on the expected direction of bias for each. It would be useful if they also specified which sub-group analyses they are going to conduct, and the rationale for these. These are only briefly mentioned on page 10, line 57.

Lines 55 to 58: why is this mixed effects model restricted to exposed women only?

Page 11

Missing data imputation

Line 5: provide justification for why key outcomes will not be imputed, provide references and possibly a bit more detail on the type of analysis that will be conducted or the rationale applied to

decide on which analysis to conduct. See, e.g., White and Royston 2009 in Statistics in Medicine, and references therein.

Ethics

Line 47: what degree of delay is “waiting for the rape case to be closed or the trial to commence” likely to introduce in data collection, and exacerbate recall bias? The choice seems sensible on the accounts described. However, it would be useful to discuss limitations, too.

Page 13

Retention challenges: the authors acknowledge substantial loss to follow up. As they note, this is not uncommon in these populations. They have put together a number of excellent strategies to retain women in the study and have created an in-built learning process into the study, with a counsellor tasked with troubleshooting new retention challenges. It is also commendable that the authors plan to study the attrition process. I am sure this will yield valuable lessons for future studies.

It would help if the authors clarified further if they use the same strategies for both cohorts, and if there is an imbalance between cohorts in the rates of attrition and how they are adjusting their retention strategies to account for this. At the moment, this is not entirely clear from the text.

Page 15

Generalisability

Lines 10 to 23: the authors correctly point out that the results reported cannot be *extrapolated to the general population*. However, it seems to me that they could make more of the fact that their controls will also be women seeking healthcare for another need (pregnancy). The nature of these two services is different and the percentage of women exposed to rape who contact services (ca. 15%, according to a previous study by some of the authors) and of pregnant women who attend ANCs (75-89% according to recent UNICEF data). However, this control cohort is better suited to this study than the general population. It would benefit the paper to include here the benefits and limitation of the socio-economic or other potential (behavioural) factors that are likely to differ systematically between the groups and could act as confounders, and how these are being tackled both at the data collection and analysis level.

Editorial and grammatical issues

In general, some plurals where you should have the singular, and some tenses are inconsistent.

References near a full stop are inconsistently placed – I would advise they all be moved to before the full-stop.

Check for typos – I have seen a few, e.g.,

P 12, line 26: initial, rather than initially

P10, para on methods, one but last sentence needs to be revised for clarity.

VERSION 1 – AUTHOR RESPONSE

Reviewer 1:

I have only a few suggestions.

One is to include questions developed and tested in an Appendix that are not existing in the literature (e.g. questions in row G and M of Table 1). This will aid in the reproducibility of the study.

We discussed the addition of the few questions that were developed for the study i.e. not yet in the existing literature. We prefer not to share it until we have done analysis of the questions and we able to report whether it works. All the other questions used are available from manuscript where the analysis are presented as well.

The second is comment on confidentiality in the context of a longitudinal study, including among contact points via whatsapp. It may be relevant to review the updated WHO guidelines on ethics and safety in intervention research on violence, which comment on follow-up visits. For whatsapp in particular, guidance on whether the phones should be personal or professional and information on norms around deleting messages should be mentioned.

We note the concern for confidentiality in the maintenance of contact in particular the use of WhatsApp and thanks for the reference to the updated guidelines. As part of the consent taking process, we discuss and list all the acceptable ways to make contact with the participant in the follow-up period. We discuss contact and update this list at each follow-up visits. We also identify women at risk for violence i.e. we ask them if their partners know about their participation in the study. This information is taken into account in the follow-up contact plan for each participant.

We do not use study phones and make contact with the participants on their personal phones.

We do not have information on norms for deleting messages and will incorporate it into our practices in the future.

We made changes under the heading Ethics and Dissemination on page14-15 . We also made changes on page 12 under Retention challenges and responses.

Finally, the authors should do a quick review of the spelling and grammar as I noticed a few places where incorrect words were used (e.g. p. 12, line 6 initially should be initial).

We apologise for the grammatical errors in the manuscript. A thorough check was done and the corrections can be seen with track changes .

Initially changed to initial on page 12

Reviewer 2

No comments to respond to

Reviewer 3:

No comments to respond to

Reviewer 4

This paper presents the protocol for a longitudinal study of the impact of rape on women's health, their use of health services and experiences of violence. It is a very interesting, valuable and very well designed study. It will fill important gaps in our knowledge in this area, and will provide much needed evidence for related research and for public health policy and interventions

I offer three types of comments and suggestions. Mostly related to the structure of the paper and study methodology.

Structure of the paper

It would be useful if the authors strengthened consistency between the research gaps they highlight in the background section, the listed outcomes, the description of the analysis they intend to carry out, and the limitations.

Below, I report a numbered list of study aims (for ease of reference) and map a possible correspondence between study aims and the gaps the authors identify in the background section. I would encourage the authors to make these parallels more explicit in the protocol, throughout the sections, and ensure they discuss aims and methods relevant to each gap, and that all gaps and methods are included for each of the listed aims.

Primary aim:

HIV acquisition + attributable burden, 24 months post rape, compared to a cohort of women with no rape.

Secondary:

- i) Incidence, attributable burden and recovery rates of phys and mental health problems that may enhance HIV risk between baseline and 24 months.
- ii) Individual, relational social and criminal justice risk factors for health problems that may be HIV risk factors and associated with symptoms persistence.
- iii) Relative importance of pathways toward HIV acquisition
- iv) Impact of rape on HIV+ survivors' ability to link to and stay in care, and sexual risk-taking behaviour;
- v) Changes in cortisol and other steroid hormone over time using hair samples, and stress response before and after trauma
- vi) Genetic and epigenetic factors in etiology of PTSD in rape exposed women
- vii) Incidence and attributable burden of CVD factors, markers of increased CVD risk, hypertension and diabetes-related renal dysfunction comparing rape-exposed and non-exposed women
- viii) Qualitative experiences of rape-exposed and non-exposed women,
 - a. & focus on motivations and experiences of the research process, retention issues and the support provided to women

Mapping between gaps/hypotheses and aims

Hypotheses and gaps in the literature Study aims

HIV may derive from rape directly primary

OR, it may become more likely via

- i) "Exposure to IPV reduces women's ability to assert themselves sexually, and determine timing and circumstances of sex, including condom use and fidelity;" Possibly refers to aim number iii, suggesting a structural equation model? Model is not explicitly mentioned in analysis
- ii) "Elevated high risk sex behavior on the part of women; including engagement in sex work or similar and having multiple partners" secondary aim number iv
- iii) "No change in women's behaviour, but drawn to men with high HIV risk." secondary aim number iv
"Further research is needed to investigate links
 - a. between rape, mental ill-health and dysfunctional hypothalamic-pituitary-adrenal (HPA) axis activity." secondary aim number i
 - b. "Research is needed to deepen our knowledge on the role of epigenetics in the development of mental health disorders post-rape." secondary aim numbers vi and v
 - c. "No study has described relationship between psychosocial stress triggered by rape exposure and development of cardio-vascular risk factors in longitudinal settings." secondary aim number vii
 - d. No gap, but mention in study challenges paragraph and elsewhere* secondary aim number viii
 - e. No gap, but mention in analysis paragraph secondary aim number ii
 - f. "Effect of rape on adherence for HIV+ women. Psychosocial factors that hinder women's adherence to care are understudied. Developing interventions that respond appropriately to the impact of rape to ensure women remain in care is important." secondary aim number iv

We discussed and considered the suggestion to make major changes to map the aims with the literature. We re-looked at this again and we believe our background section provides a clear overview of the problem which are largely the knowledge gaps in the literature. We think the study aims are linked to what we present in the background. Some of our study aims such as the qualitative study was added later in response to our experiences in implementing the study rather than addressing our original study aims. Other aims as explained lower down are PhD studies that emerged as collaborations were formed and these sub studies will have their own proposals developed.

*Explaining retention challenges in this (and similar, e.g., IPV survivors?) populations is missing from the background section, but qualitative research to explore these is introduced in the aims (secondary aim number viii) and explained as a later addition to understand challenges (p12, line 28). I would suggest the authors include considerations on these challenges and on the paucity of specific research. Incidentally, there is some on IPV by Gondolf, as the authors are almost certainly aware. As an aide memoir, a lot of it does focus on perpetrators, but see Gondolf and Deemer 2004: though US-focused, this paper may offer some interesting insights.

It would be useful if in the analysis section the authors also explained that the study of retention challenges is to be conducted using qualitative methods (only?), and gave some information on which methods they intend to use (data collection methods, sample size and timing of interviews, type of analysis, limitations etc.) throughout the paper.

Relatedly, discussion of evidence gaps in women's experiences (to be investigated with qualitative data – see secondary aim viii) is also not included. Would the authors consider inserting it, similarly to retention issues, to contextualise the qualitative component of the study?

We provide a detail section on retention challenges in this study as part of the Methodology/Design Section on page 12-13. In the Background section we only locate the state of knowledge on logistical issues in doing longitudinal studies with GBV populations (page 5). We also refer to it in the Lessons Learnt and Limitations - see page 13-14. We believe the detail experiences we present on doing longitudinal follow-up studies among GBV participants is one of the strengths of the paper.

We added the qualitative study in response to our experiences in retention after the start of the study as stated in the manuscript. Page 13 under STUDY PROGRESS AND CHALLENGES

The paper by Goldolf and Deemer as suggested were read and challenges on retaining participants (batterers and their new partners) using telephone calls are presented in this paper. Postal interviews were used which requires a very different retention strategy to asking participants to return to a research clinic, however there are similarities in terms of maintaining telephonic contact. We found useful lessons were from GBV intervention research and the literature was trawled again to look for papers that discuss retention and tracking in GBV studies. However, retention strategies are often not discussed in detail. We believe our detail discussion on our experiences on retention to date are adequate for this proposal paper and we hope this contribution is of value for those planning similar studies.

The request to share more detail on the qualitative study was discussed by authors. We are concerned that this will create confusion with the cohort study design. Many of the sub studies are being developed by PhD students as individual research proposals and although we have the methodological detail of the qualitative study we would have to also provide same level of detail for all the other sub-studies and this will not be possible. Two of the sub studies are preparing their proposals for publication in peer review journals and will use this paper manuscript if published as a reference. We hope this is an acceptable explanation.

Methodological comments

The following paragraphs report comments on the methodological issues that arise reading the paper, so they mostly follow the order in which information is presented in the paper.

Page 7

Study site/sites: the paper would benefit from a clearer description of the sites. The “Setting” section (page 7, lines 37 to 45) describes the general settings where participants are recruited. However, it does not provide information on the total number of centres recruitment happens at, nor of the urbanicity of the sites. Further, the role of and rationale for the research site is not clearly introduced in this paragraph. Later in the paper, there are mentions of participants being transported to the site for tests. It would aid with clarity if the role of the research site and women’s visits to the site were introduced in the “setting” section

More generally, I wonder if the authors would consider providing a bit more detail on the study processes to help the reader understand the study a bit better. The paper contains detailed information on which tests are done at which points in time, and which questions are asked at each point in time, and measurement is clearly discussed. However, some more explanation of the context would help the reader gain a better understanding of the study process. For example, the authors mention “these four sites” on page 14, lines 29-30. However, these have not been introduced beforehand in the text and it is unclear what the authors are referring to. They do mention having had to increase the number of “study sites” (page 12, line 27), but this is not necessarily related to the following remark on page 14. Also, there is an explicit mention of (one of) the site(s) (page 14, line 16 – is this the research site? Is this where the research clinic is? Unclear), which the authors might consider turning into a more generic indication of locality to further protect participants’ privacy. They may wish to consider further specifying the function of this site in relation to the others to allow readers to gain a better understanding of the study process, in place of providing its name. Similarly, see the mention of “RICE” on page 10: as noted below, this is not introduced anywhere in the paper. A short explanation of the flow of participants through the study and relevant research actors could prove useful and could be included in this section. For example, in addition to the recruitment and retention strategies described, and of the number and geographical location of the recruitment sites, the authors could also include some information on where and how the various follow up visits are done, and whether this changes between groups; the role of the research site (same as study site?), and who is involved in taking measurements. Perhaps a figure/diagram would be sufficient?

We apologise for the confusion between study recruitment sites and the site where the study clinic is based. We made substantial changes in the Settings section (page 7) and wherever else we refer to the sites.

We explain the rationale for the recruitment sites.

We apologise for omitting to explain the study acronym. We introduce the shortened name used for the study (RICE) at the start of the manuscript on page 4

We do provide detail on who does the interviews and assessments on page 9 under Assessments of Exposure and Outcome Measures

We added a flow diagram as Figure 1 to assist with understanding the study flow and processes such as the follow-up visits and tests.

We explain that both groups have the same number of follow-up visits and same assessments (page 8 under Assessments of Exposure and Outcome Measures).

See also:

Page 10

Lines 33-34: RICE staff mentioned for first time. RICE not been previously introduced in paper. RICE study is then mentioned several times, but it is not made clear what/who RICE is. Would the authors consider spending a few words clarifying who RICE are, and what their role in the project is? This becomes indirectly apparent later, at least in part. It would be good to have an introductory succinct description of their role.

And

RICE acronym clarified see page 4.

Page 14

Line 30 “these four sites” unclear. These sites have not been introduced before.

Clarified under Setting page 7.

Finally, the authors may want to explain the reason for their choice of recruitment centres for exposed women. This may not be clear to readers not familiar with the South African context, because they may not be aware of the role of the TCCs and the challenges rape survivors face when reporting at police stations. If similar data exist, it would also be useful to provide an indication of the percentage of rapes reported to TCCs versus other services, to get a sense of how representative of rape survivors who report to services the sample may be.

See changes to the Settings section where the TCCs are discussed in greater detail (page 7)

We do not have data on the percentage of rapes reported to the TCC (or other services) and cannot make such comparisons. Such service level data is not in the public domain and comparisons can therefore not be made. When we planned the study, we were given data for the estimation of the accrual period and we agreed not to share this data.

Line 16: tablets indirectly introduced?

Corrected

55: unclear – is PTSD not collected at baseline? Why not? How are acute stress reactions measured?

We do assessments of PTSD at baseline and at all follow-up visits. We clarified this statement to indicate that PTSD for the rape incident cannot be made at baseline as PTSD is time bound.

(Inclusion and Exclusion Criteria - page 7)

Page 8

Inclusion & exclusion criteria, and confounding

Line 3:

I like the strategy of only including women engaged in services this will reduce the impact of behavioural and socio-economic confounders. Matching on age also very good.

However, control women may be different from exposed women on other accounts such as, e.g., socio-economic factors like educational attainment, labour force participation, etc. I would encourage the authors to discuss these and/or similar confounding variables, and discuss how they are tackling this challenge. Are they collecting data on relevant confounders, if they are not stratifying by these at the data collection stage? How do they think confounders may bias study results (in which direction), and what are they doing to account for them? In the limitations section, the authors are also encouraged to discuss how bias from confounders may affect study results and what analytical strategies they are putting in place to mitigate this.

We are confident that the two groups are very similar. We purposefully recruit the un-exposed from health services in the same areas of the rape services. We collect data on relevant confounders which have been identified in the literature. These are socio-economic variables including education, social grants, employment, housing etc. We also collect information on known confounders for HIV acquisition and for sexual violence such as sexual behaviour, alcohol and drug use etc. These are listed in Table 1. In addition, we collect data on childhood trauma and exposure to other traumas as these are known confounders for many of mental health outcomes we are measuring. We will consider all relevant confounders in all the analysis.

Our study will do many analysis and we can never be able to provide a detail plan for the numerous analysis we will do. It will be an iterative process and we will be testing confounders as part of for all of these analysis. We added incorporated issues related to confounders on page 10 and 11.

Line 5: Related to the above point, excluding women with lifetime exposure to sexual violence is

justifiable if the authors want to see the effect of rape on HIV incidence on women who are exposed to (recent) violence. However, this choice is likely to reduce the comparability between the two groups, as the women in the exposure group are likely to have been exposed to lifetime violence prior to the incident that triggered their contact with the services. Could the authors elaborate further on this?

We are aware that prior exposure is very likely in our setting and we measure this through the Childhood Trauma and Life Events Checklist scales (Table 1) to control for in analysis as discussed above.

In addition, could the authors mention how exposure to sexual violence after recruitment may affect participation in the study for women in the control group?

If women from our control group seek services post rape at one of the recruitment sites and meet eligibility i.e. report the rape within 20 days of the incident she will switch groups and will be recruited as a new baseline participant in the exposure group. If women from the control group however report the rape late we will not switch her to the expose group. We do collect data on all types of violence experiences in the period between visits and will have to separate this group in the analysis. This was added to page 9.

Line 7: "Other exclusion criteria, as for rape exposed women, are applied." Unclear. Could the authors clarify if all other criteria or a subset are followed?

We clarified exclusion in the non-exposed group. Page 7.

Measurement

Line 40: AUDIT-C would be sufficient to determine alcohol misuse. Authors may wish to consider using this three-time version of the measure instead of the 10-item AUDIT measure to reduce interviewee burden. The AUDIT-C has been extensively used in South Africa, and has generally shown good psychometric properties and high correlation with the AUDIT measure (see, e.g., Hartley et al. 2011 in Reproductive Health)

Thank you for this advice. We are more than half way through out accrual and we are not keen to make changes to our assessments.

Page 10

Analysis

"Data management and analysis" section: I would encourage the authors to review this section. Regarding the analysis, the authors say they will estimate Cox proportional hazards models, and mixed effects and random effects models for different types of analyses. It would improve the paper's transparency if they mentioned which covariates they are considering including in these models. For a cohort study, in all of these models, confounders due to imbalance and variation in covariates will have an effect on the estimates. It is therefore very important that the authors show they are aware of this, and have been thinking about which covariates should be controlled for in the model, given that only age was included in the matching/stratifying strategies. I would encourage the authors to discuss which confounders they are measuring, and provide indication on the expected direction of bias for each. It would be useful if they also specified which sub-group analyses they are going to conduct, and the rationale for these. These are only briefly mentioned on page 10, line 57.

Lines 55 to 58: why is this mixed effects model restricted to exposed women only?

See above the discussion on confounders.

Page 11

Missing data imputation

Line 5: provide justification for why key outcomes will not be imputed, provide references and possibly a bit more detail on the type of analysis that will be conducted or the rationale applied to decide on

which analysis to conduct. See, e.g., White and Royston 2009 in Statistics in Medicine, and references therein.

See rationale added on pages 10 and 11.

Ethics

Line 47: what degree of delay is “waiting for the rape case to be closed or the trial to commence” likely to introduce in data collection, and exacerbate recall bias? The choice seems sensible on the accounts described. However, it would be useful to discuss limitations, too.

We have agreed to this with the National Prosecuting Authority not ask question related to the case until case is closed or trial commences. Our study covers a definite period of 2 years and even longer for those who entered the study early and we have a good chance to collect data on the legal outcome. We do not think there will be an issue of recall bias as the outcome of the legal process is of great important to most people. We also expect to have many cases closed either by the survivors or by the investigating team as shown in rape attrition research in South Africa. We added this to the Ethics section on page 17.

Page 13

Retention challenges: the authors acknowledge substantial loss to follow up. As they note, this is not uncommon in these populations. They have put together a number of excellent strategies to retain women in the study and have created an in-built learning process into the study, with a counsellor tasked with troubleshooting new retention challenges. It is also commendable that the authors plan to study the attrition process. I am sure this will yield valuable lessons for future studies.

It would help if the authors clarified further if they use the same strategies for both cohorts, and if there is an imbalance between cohorts in the rates of attrition and how they are adjusting their retention strategies to account for this. At the moment, this is not entirely clear from the text.

We find retention challenges in both groups and apply the same efforts and interventions for both. We will assess differences in loss to follow- up in the groups at a later stage in the study. We do indicate that the challenges are found across the two groups page 14.

Page 15

Generalisability

Lines 10 to 23: the authors correctly point out that the results reported cannot be extrapolated to the general population. However, it seems to me that they could make more of the fact that their controls will also be women seeking healthcare for another need (pregnancy). The nature of these two services is different and the percentage of women exposed to rape who contact services (ca. 15%, according to a previous study by some of the authors) and of pregnant women who attend ANCs (75-89% according to recent UNICEF data). However, this control cohort is better suited to this study than the general population. It would benefit the paper to include here the benefits and limitation of the socio-economic or other potential (behavioural) factors that are likely to differ systematically between the groups and could act as confounders, and how these are being tackled both at the data collection and analysis level.

We have explained the issue of confounding as explained above. It is unlikely that the control group differ markedly from the exposed group and we are measuring the factors that could increase the risk for both HIV and victimisation of sexual violence including mental health outcomes which could place women at increased risk for both outcomes.

Editorial and grammatical issues

In general, some plurals where you should have the singular, and some tenses are inconsistent.

References near a full stop are inconsistently placed – I would advise they all be moved to before the full-stop.

Check for typos – I have seen a few, e.g.,

P 12, line 26: initial, rather than initially

P10, para on methods, one but last sentence needs to be revised for clarity.

Thank you for identifying the above. We have done a review and made corrections and hope we have identified all the grammatical mistakes.

VERSION 2 - REVIEW

REVIEWER	Giulia Ferrari London School of Hygiene and Tropical Medicine
REVIEW RETURNED	11-Jun-2017

GENERAL COMMENTS	The paper's clarity is now much improved, further highlighting the rigour and thoughtfulness of the study. I would encourage the authors to have the paper closely proof-edited, as quite a few typos and grammatical imprecisions remain. I wish the authors all the best in continuing and concluding this very valuable study.
---

VERSION 2 – AUTHOR RESPONSE

I have sought assistance from a copy-editor to review the manuscript for grammatical mistakes.

Changes have been made throughout the manuscript where required.